# Stereotactic Radiosurgery for Benign Cavernous Sinus Meningiomas: A Multicentre Study and Review of the Literature

**DOI:** 10.3390/cancers14164047

**Published:** 2022-08-22

**Authors:** Antonio Santacroce, Constantin Tuleasca, Roman Liščák, Enrico Motti, Christer Lindquist, Matthias Radatz, Brigitte Gatterbauer, Bodo E. Lippitz, Roberto Martínez Álvarez, Nuria Martínez Moreno, Marcel A. Kamp, Bente Sandvei Skeie, Stephanie Schipmann, Michele Longhi, Frank Unger, Ian Sabin, Thomas Mindermann, Otto Bundschuh, Gerhard A. Horstmann, A.T.C. J. van Eck, Maja Walier, Manfred Berres, Makoto Nakamura, Hans Jakob Steiger, Daniel Hänggi, Thomas Fortmann, Samer Zawy Alsofy, Jean Régis, Christian Ewelt

**Affiliations:** 1Department of Neurosurgery, St. Barbara-Klinik Hamm-Heessen, 59073 Hamm, Germany; 2Department of Medicine, Faculty of Health, Witten/Herdecke University, 58455 Witten, Germany; 3Neurosurgery Service and Gamma Knife Center, Lausanne University Hospital (CHUV), 1011 Lausanne, Switzerland; 4Faculté de Biologie et de Médecine (FBM), Université de Lausanne (Unil), 1005 Lausanne, Switzerland; 5Signal Processing Laboratory (LTS 5), Swiss Federal Institute of Technology (EPFL), 1015 Lausanne, Switzerland; 6Faculté de Médecine, Sorbonné Université, 70513 Paris, France; 7Assisstance Publique-Hôpitaux de Paris, Hôpitaux Universitaires Paris Sud, Centre Hospitalier Universitaire Bicêtre, Service de Neurochirurgie, 94270 Le Kremlin-Bicêtre, France; 8Department of Stereotactic and Radiation Neurosurgery, Na Homolce Hospital, 15000 Prague, Czech Republic; 9Dipartimento di Neuroscienze, Neurochirurgia, Università degli Studi di Milano, 20122 Milano, Italy; Villa Maria Cecilia Hospital, 48033 Cotignola, Italy; 10Gamma Knife Centre, Bupa Cromwell Hospital, London SW5 0TU, UK; 11National Centre for Stereotactic Radiosurgery, Royal Hallamshire Hospital, Sheffield S10 2JF, UK; 12Department of Neurosurgery, Medical University of Vienna, 1090 Vienna, Austria; 13Interdisciplinary Centre for Radiosurgery (ICERA), Radiological Alliance, 22767 Hamburg, Germany; 14Department of Radiosurgery, Rúber International Hospital, 28034 Madrid, Spain; 15Department of Neurosurgery, Jena University Hospital, Friedrich-Schiller-University Jena, 07747 Jena, Germany; 16Department of Neurosurgery, Haukeland University Hospital, 5021 Bergen, Norway; 17Unit of Radiosurgery and Stereotactic Neurosurgery, Department of Neurosciences, Azienda Ospedaliera Universitaria, 37126 Verona, Italy; 18Department of Neurosurgery, Medical University Graz, 8036 Graz, Austria; 19Gamma Knife Unit, Wellington Hospital (Platinum Medical Centre), London NW8 7JA, UK; 20Gamma Knife Center Zurich, Klinik Im Park Hirslanden, 8002 Zurich, Switzerland; 21Gamma Knife Zentrum Hannover, 30167 Hannover, Germany; 22Gamma Knife Zentrum Krefeld, 47805 Krefeld, Germany; 23Institute of Medical Biometry, Epidemiology and Informatics, University Medical Center of Mainz, Langenbeckstrasse 1, 55131 Mainz, Germany; 24Department of Mathematics and Technology, University of Applied Sciences Koblenz, Joseph-Rovan-Allee 2, 53424 Remagen, Germany; 25Department of Neurosurgery, Academic Hospital Köln-Merheim, 51058 Köln, Germany; 26Department of Neurosurgery, Heinrich-Heine-Universität Düsseldorf, 40225 Düsseldorf, Germany; 27Service de Neurochirurgie Fonctionnelle et Stereotaxique, Hôpital D’adulte de la Timone, 13354 Marseille, France; 28Department of Neurosurgery, University Hospital Münster, Albert-Schweitzer-Campus 1, A1, 48149 Munster, Germany

**Keywords:** meningioma, cavernous sinus, stereotactic radiosurgery, Gamma Knife, multicentre study

## Abstract

**Simple Summary:**

Meningiomas are the most common tumours of the central nervous system (CNS). Despite their benign histology, proximity to critical neurovascular structures may lead to significant morbidity with tumour growth. This is the case for cavernous sinus meningiomas (CSMs), as their growth may surround critical neuro-vascular structures and cause significant morbidity. Radical microsurgical resection carries a high risk of additional neurological deficits, as well as the risk of death. Current management of these tumours, where treatment is indicated, has moved away from radical surgery towards radiotherapy/radiosurgery. This is particularly the case for patients who have residual or recurring tumours after previous surgery. There are many reports that describe the effectiveness of using stereotactic radiosurgery (SRS) for CSMs; however, large cohort analyses are lacking. This multicentre analysis reports the outcome data of over 1000 patients with CSMs who were treated with SRS. SRS shows a high local tumour control rate with few complications. These results agree with previous reports in the literature. SRS is a valuable primary or adjuvant treatment option for CSMs.

**Abstract:**

Cavernous sinus meningiomas (CSMs) remain a surgical challenge due to the intimate involvement of their contained nerves and blood vessels. Stereotactic radiosurgery (SRS) is a safe and effective minimally invasive alternative for the treatment of small- to medium-sized CSMs. Objective: To assess the medium- to long-term outcomes of SRS for CSMs with respect to tumour growth, prevention of further neurological deterioration and improvement of existing neurological deficits. This multicentric study included data from 15 European institutions. We performed a retrospective observational analysis of 1222 consecutive patients harbouring 1272 benign CSMs. All were treated with Gamma Knife stereotactic radiosurgery (SRS). Clinical and imaging data were retrieved from each centre and entered into a common database. All tumours with imaging follow-up of less than 24 months were excluded. Detailed results from 945 meningiomas (86%) were then analysed. Clinical neurological outcomes were available for 1042 patients (85%). Median imaging follow-up was 67 months (mean 73.4, range 24–233). Median tumour volume was 6.2 cc (+/−7), and the median marginal dose was 14 Gy (+/−3). The post-treatment tumour volume decreased in 549 (58.1%), remained stable in 336 (35.6%) and increased in only 60 lesions (6.3%), yielding a local tumour control rate of 93.7%. Only 27 (2.8%) of the 60 enlarging tumours required further treatment. Five- and ten-year actuarial progression-free survival (PFS) rates were 96.7% and 90.1%, respectively. Tumour control rates were higher for women than men (*p* = 0.0031), and also for solitary sporadic meningiomas (*p* = 0.0201). There was no statistically significant difference in outcome for imaging-defined meningiomas when compared with histologically proven WHO Grade-I meningiomas (*p* = 0.1212). Median clinical follow up was 61 months (mean 64, range 6–233). Permanent morbidity occurred in 5.9% of cases at last follow-up. Stereotactic radiosurgery is a safe and effective method for treating benign CSM in the medium term to long term.

## 1. Introduction

The management of cavernous sinus lesions remains a major challenge, despite the latest advances in microsurgery and operating room facilities. The ultimate goal of microsurgery is complete tumor resection, with minimal morbidity and no mortality. Benign meningiomas are the most common lesions arising within the cavernous sinus. Until the introduction of MRI and CT, cavernous sinus meningiomas (CSMs) were often diagnosed late, and the prognosis was considered poor [1,2,3,4]. Despite results published from specialised centres [5,6,7,8], surgery within the cavernous sinus often increases neurological deficits and carries a risk of death [8,9,10,11,12,13]. This is due to the complex anatomy of CSMs, including a segment of the internal carotid artery (ICA); the sympathetic plexus and cranial nerves (CN) III, IV, Va, Vb and VI; and the high blood flow within the venous plexus [14,15,16,17]. For small- and medium-sized CSMs, stereotactic radiosurgery (SRS) is an alternative that provides high local tumour control and significant improvement in CN deficits [1,18,19]. Large tumours, however, with extra-cavernous extension or compression of the optic pathways, are generally treated currently using subtotal microsurgical resection, followed by adjuvant SRS or simple observation over time [1,2,4,20,21,22,23,24,25,26,27,28,29,30,31,32,33,34,35,36]. In order to enable informed decisions about treatment options, this series documents local control rates and morbidity of a large series of CSMs treated by SRS.

Subject to analysis is a subgroup of patients diagnosed with CSM and treated with SRS in 15 European Gamma Knife centres.

The aim of the study is to assess the long-term efficacy and safety of SRS for benign CSMs with respect to tumor growth and prevention of associated neurological deterioration. Medium- to long-term outcomes have been widely reported, but no large multicentre series studies with long-term follow-up have been published.

## 2. Materials and Methods

### 2.1. Conceptualization

This study was initiated by the European Leksell Gamma Knife Society. The Gamma Knife Centre in Krefeld, Germany, defined the study protocol and coordinated the data retrieval. The study was approved by the local ethical committee of the Heinrich Heine University of Düsseldorf (ethical approval number 4002) [24].

### 2.2. Multicentric Database

We initially defined a database for data retrieval and further analysis, which was then divided into 2 sections. The first section reported about clinical and neurological statuses before SRS, treatment parameters, and tumour features. The second section included imaging and clinical neurological follow-up over after treatment [24].

All centres contributed a minimum of 50 meningiomas, the first tumour in each centre having been treated before the year 2000. An agreement was made with all treating physicians that cohort details, diagnostic, treatment, and follow-up protocols and other necessary information would be made accessible to the first author as required. After the protocol was defined, data were retrieved [24].

### 2.3. Data Retrieval

Data were retrieved by the first author and entered into a structured database, including the clinical neurological examination before treatment and at last follow-up, treatment parameters, and tumour characteristics before treatment and at the last available follow-up. For the current sub-analysis, we included all patients harbouring a cavernous sinus meningioma [24].

All treatment protocols and follow-up material were retrospectively reviewed personally by the first author [24]. Each centre was visited at least once over a period of 18 months; data were drawn from imaging studies, clinical notes and the local databases in all 15 centres under the supervision of the referring physician [24].

The data were then unified into a single database and completed in accordance with the defined model. Each referring physician received a copy of the local database adapted to the model of the mean database in order to monitor the quality of retrieval [24]. The first author had free access to clinical and personal data of patients. No patient names or other forms of personal identification were entered into the database [24]. A single unique identification number was attributed to each centre, and another unique number was allocated to each individual patient in order to allow for a centre-stratified analysis [24]. No member had a copy of the main database, as such a copy was generated solely for the purpose of statistical analysis [24].

### 2.4. Cohort Description

Between May 1988 and November 2003, 1206 patients harbouring 1256 CSMs were treated in these participating centres. Sixteen patients undergoing repeat SRS for 16 enlarging tumours were considered to be new cases. A cohort of 1222 patients harbouring 1272 CSMs was finally reviewed (Figure 1). 

Of 1222 patients treated, 150 (12%) were lost to follow-up. There were further detailed results for 1116 tumours (87.7%), which were collected in the follow-up section of the database. A neurological examination was available for 1024 patients (85%).

All tumours were either histologically confirmed as WHO grade I, or were considered benign from radiological appearances. A total of 677 patients harbouring 703 tumours underwent SRS in definitive setting. The remaining 545 patients were treated for WHO Grade 1 meningiomas (569 tumours).

SRS was performed using the Leksell Gamma Knife (Elekta Instruments AB, Sweden), and the most current planning software at the time of treatment.

Sporadic lesion was defined as a single lesion treated in the cavernous sinus in a single patient. Multiple lesions were defined as two or more meningiomas non-recurring from the same surgical field, with one of the targets arising from the cavernous sinus. Within the second group, we further identified and subdivided lesions into those that were associated with type II neurofibromatosis (NF2) and those patients with frank meningiomatosis [24].

Indications for SRS were tumours with no histological confirmation, remnant or recurrence following microsurgical resection, with maximum major tumour diameters of less than 3 cm and with acceptable dose delivery to adjacent eloquent structures. Specifically, in patients with intact vision, the marginal dose to any part of the optic apparatus was restricted to 10 Gy [37]. Tumours with a clear arising origin from the cavernous sinus loge with infiltration of the surrounding tentorium, i.e., meningeal tissue in middle cranial fossa were defined as cavernous sinus meningioma.

### 2.5. Endpoints

The primary endpoint was to estimate progression-free survival rate (PFS) after treatment, and to assess the influence of several variables on this outcome. Imaging tumour control was defined as a stability or shrinkage of at least 10% of the target volume assessed by volumetric measurement. A tumour enlargement of 10% or more compared to the target volume was defined as progress.

The secondary endpoint was to confirm SRS safety by establishing clinical neurological stability and complication rates (either clinical and/or radiation adverse events) after SRS [24].

### 2.6. Imaging Follow-Up

Serial imaging (MRI or CT when MRI was contraindicated) was performed at various times according to each centre, and the results were collated in the follow-up section of the main database. Qualitative and quantitative evaluations of tumour size were performed on each examination by the local staff and independently reviewed by the first author, who also compared reported outcomes. Tumour volume on each scan was compared with the tumour volume before SRS. If the imaging material was digitalized, a volumetric measurement was performed with dedicated software. When imaging was available only on celluloid film, the comparison was made by measuring and comparing the 3 major diameters. Shrinkage or enlargement was defined as an imaging-assessed change in tumour volume of at least 10% determined (as described above) either by direct volumetric measurement or by calculation bas ed on dimensions (Table 1).

### 2.7. Neurological Assessment and Clinical Follow-Up

Before treatment, all patients underwent neurological examination, which was available for all patients. Clinical data were standardized according to the following uniform classification [24]: 0 = no neurological deficit; 1 = mild or intermittent neurological deficit; 2 = persisting neurological deficit but not affecting performance in daily life; and 3 = permanent/severe neurological deficit affecting performance in daily life. Epileptic seizures were classified as follows: 0 = no seizure activity; 1 = partial seizures; and 2 = generalized seizures. Each epileptic category was subdivided into temporary and permanent subgroups. Clinical and imaging follow-up were performed thereafter (Table 2).

Any neurological deficit newly reported or reported to be worsening after SRS was carefully evaluated and defined as a complication [24].

Complications were then further divided into temporary and permanent subgroups. The latest clinical follow-up is reported separately from the imaging follow-up. In the event of death, date of death was considered to be the date of latest clinical follow-up, and the cause of death was classified as treatment related (treatment), tumour related (meningioma), unrelated (other) or unclear (unknown). Complications are described separately in patients who died of unclear causes. Patients without clinical control were defined as lost to follow-up, and patients without imaging follow-up were defined as lost to imaging follow-up. They are not included in the statistical analysis [24].

### 2.8. Survey of Patients

Thus, all patients had been treated at least 5 years before this retrieval visit, but it does not follow that all tumours had an effective follow-up of 5 years [24]. A survey of patients lost to follow-up was conducted at 5 years in 11 centers by sending a follow-up letter to the patient or referring physician [24]. Patients were asked to undergo further imaging and neurological evaluation at the treating centre, and physicians were asked to provide updates based on this imaging and to plan any necessary further imaging accordingly [24]. They were then requested to return such results to the treating centre; these results were then subsequently collected by the local research fellow and finally reviewed by the first author [24].

### 2.9. Statistics

Statistics were performed as detailed in our previous publication [24]. Analysis was performed to evaluate the imaging outcome. Time to enlargement of target volume was estimated with the Kaplan–Meier method, was defined by date of the first scan to show tumour growth, rather than by time to effective regrowth [24]. All target volumes with less than 24 months imaging follow-up were excluded. Independent variables were evaluated separately in order to estimate their influence on imaging outcome. Univariate comparisons between non-continuous variables (histology, patients’ gender, multiple vs. sporadic lesions) were performed with the log rank test [24]. Univariate analysis for continuous variables (prescription margin dose, maximum dose, age, volume and isodose distribution) was performed by applying the Cox proportional hazard model [24]. Multivariate analysis was performed using stepwise Cox regression [24]. For the purposes of this analysis, we used only those variables that had been demonstrated by the prior univariate analysis to be significant [24]. We included patients with more than one meningioma, with frank meningiomatosis and neurofibromatosis type 2 [24]. These cases were analysed separately and singularly compared with sporadic cases, in order to evaluate a statistically significant difference in outcome with respect to PFS rate. Statistical analyses were performed with the SAS software package (SAS for Windows, version 9.1; SAS Institute Inc., Cary, NC, USA) [24].

## 3. Results

### 3.1. Local Tumour Control Assessment

Median imaging follow-up was 67 months (mean 73.4, range 24–333). A total of 171 tumours, for whom follow-up was less than 24 months, were excluded. From this cohort we extracted data for follow-up of greater than 5 years (595), 7.5 years (261) and 10 years (103). We found that 549 (58%) meningiomas had regressed (533 patients) and 336 (35.6%) tumours had remained unchanged (325 patients) during follow-up, giving a tumour control rate of 93.6%. Tumour progression occurred in 60 (6.4%) lesions, after a median interval of 57.7 months (mean 62 months). Of the former, 16 patients with 16 enlarging tumours underwent further treatment. Conventional radiotherapy or repeat SRS was performed in four of these patients, and eight underwent surgery. The remaining four tumours had not required further treatment by the time of last follow-up. The Kaplan–Meier estimations of progression-free survival rate (PFS) at 5 years, 7.5 years and 10 years showed control rates of 96.7%, 92.7% and 90.1%, respectively, for the overall data. We observed PFS rates of 97.2%, 90.1% and 87.9% for those tumours initially treated with microsurgery, versus 97.2%, 95.3% and 92.3%, respectively, for tumours without histological confirmation (Figure 2, Figure 3 and Figure 4). Tumour control was better in females than males (*p* = 0.0079), and for patients with single rather than multiple meningiomas (*p* = 0.0201) (Table 3, Table 4 and Table 5). A statistically significant difference in tumour control between centres was also observed (*p* = 0.0160). A representative case is shown in Figure 5.

### 3.2. Neurological Assessment

Neurological assessment was available for 1042 (85.2%) patients, with follow-up ranging from 6 to 233 months. Clinical improvement was reported in 460 (44.2%) patients at a median follow-up of 62 months (mean 65.6), and complete resolution of symptoms was reported in 241 (23.2%) cases. In particular, we observed an improvement in the symptoms related to the cranial nerves in 27.6% of cases with pre-existing deficits (288 cases), and complete resolution of symptoms in 144 cases (13.8%). Of note, we observed neurological improvement of 52.8% in patients undergoing primary SRS and 28.6% in post-operative patients with WHO Grade I CSMs. Complete resolution of symptoms in these two groups was seen in 26.2% and 14.1% of patients, respectively (Table 6).

Complications were observed after SRS in 113 (11.3%) patients, as detailed in Table 7. Transient morbidity rates were 5.4% and permanent were 5.9%. No radiation-induced tumours were seen, but in two patients that were initially treated with SRS and subsequently operated on for post-treatment tumour enlargement, histology revealed a craniopharyngioma WHO Grade 1 and an anaplastic meningioma WHO Grade 3, respectively.

## 4. Discussion

### 4.1. Our Results

To our knowledge, the present study includes the largest consecutive series of patients treated for CSM with SRS. The median imaging follow-up was over five years. Five- and ten-year actuarial PFS rates were 96.6% and 90%, respectively, which are comparable to previous publications [1,2,3,4,18,28,31,34,38].

Tumour control was higher for solitary tumours (*p* = 0.0097). Although there is no data about SRS for CSMs in multiple lesions or NF2, the sub-analysis confirms our previous results [24].

In our previous analysis we found, contrary to other series [18,39], that previous surgery significantly reduced tumour control [24,39]. This was not confirmed in the current sub-analysis (*p* = 0.1212) (Table 4), thus implying that better dose planning was possible despite distortion of anatomy by prior surgery [1].

We reported poorer control in males in our previous paper (*p* = 0.0031), and although the reason is not known, hormonal influences may be a factor [8,24].

Although dose concepts and treatment planning are quite standardized for SRS of CSMs [1,2,28,38,40], a statistically significant difference between centres seems to be still an issue, thus confirming our previous finding [24].

### 4.2. CSM and Microsurgical Management

*CSMs* have an estimated incidence of 0.5 per 100,000 [5,6,7,41,42], and their management remains controversial. The complex anatomy of this area, with its contained cranial nerves and carotid siphon makes radical surgical resection challenging, as it carries a high risk of neurological deficits [18]. Dolenc [43,44,45], whose work led to a better appreciation of the surgical anatomy of the cavernous sinus, was an early pioneer of microsurgery; however, as patients have become better informed and less tolerant of surgical morbidity, enthusiasm has waned. Subtotal microsurgical resection leads to fewer complications but is associated with high recurrence rates in the medium to long term. Sindou et al. [12] published a series of 100 patients with CSMs who showed extra-cavernous extensions in follow-ups ranging from 3 to 20 years (mean 8.3 years). Gross-total removal of both the extra- and intra-cavernous portions was achieved in 12 patients (12%); removal of the extra-cavernous portions with only a partial resection of the intra-cavernous portion was achieved in 28 patients (28%); and removal only of the extra-cavernous portions was achieved in 60 patients (60%). Significant additional deficits were reported after surgery, with a higher rate of complications when resection was performed within the cavernous sinus. Five patients died (5%). Tumour regrowth was observed only in 11 of the 82 surviving patients with subtotal removal, however this recurrence rate may increase with additional follow-up in these slowly growing tumours, and with better quality post-operative imaging. The authors concluded that there was no oncological benefit to performing radical resection on the intra-cavernous tumour. Heth et al. [42] reported a series of 163 patients. Total removal was achieved in only 71 cases (44%) and tumour recurrence was observed in 7%. In the cases with partial resection, further growth was seen in 57%.

### 4.3. Radiation Therapy for CSMs

The role of stereotactic radiation therapy delivered in a single session (SRS) for the management of intracranial benign meningiomas has been well established, as either a primary or adjuvant treatment [1,2,4,18,20,24,25,27,28,29,30,31,32,33,34,40,46]. There is also evidence for the use of stereotactic dose fractionated radiotherapy (SFRT) [25,46,47,48,49,50,51,52]. In a recent systematic review of the role of SRS and SFRT for treating CSM using different platforms, PFS rate at 5 and 10 years after GK-SRS was 93.6% and 87% respectively. PFS rate after Linac based SRS was 95.6% at 5 years and 87.4% at 10 years. PFS rate at 5 and 10 years after Linac based SFRT were 97.4% and 95.5% respectively [38]. However, the indication for treatment and the method of defining target volumes were not uniform. Over 70% of the patient data were from gamma knife centres, with a longer median follow-up, whereas the 10-year statistical results for Linac SRS and SFRT were generally extrapolated from actuarial tables and may have overestimated the imaging outcome [8,10,53,54].

The recent International Stereotactic Radiosurgery Society Guidelines suggest that the literature is limited to level III evidence with respect to the outcomes of SRS for CSMs; however, SRS appears to have more benefits than risks for patients with CSMs [55].

The recent report of the North American Gamma Knife Consortium [40] stated that 61% of patients showed tumour shrinkage, 2% were unchanged, and 15% showed increases in tumour volume after a follow-up period of 101 months. Actuarial tumour control rates at the 5-, 10-, and 15-year follow-ups were 92%, 84%, and 75%, respectively. Of the 120 patients who had undergone SRS as a primary treatment, tumour progression was observed in 14 (11.7%) patients at a median of 48.9 months (range 4.8–120.0 months). The actuarial tumour control rates were 98%, 93%, 85%, and 85% at the 1-, 5-, 10-, and 15-year follow-ups post-SRS, respectively. Fifteen (7.5%) patients experienced permanent CN deficits without evidence of tumour progression at a median onset of 9 months (range 2.3–85 months) after SRS. Patients with larger tumour volumes (≥10 cm^3^) were more likely to develop permanent CN complications. Three patients (1.5%) developed delayed pituitary dysfunction after SRS.

It is now recognised that the higher radiation dose does not significantly influence PFS rate. Kondziolka et al. reported no improvement in PFS with single fraction doses greater than 15 Gy [56]. Lee and co-authors [4] reported the outcomes of 159 patients with CSMs treated either with adjuvant or primary SRS and found no statistical difference. More recently, Pollock et al. [1] reported 115 cases of CSMs treated with the Gamma Knife between 1990 and 2008. Grade II and III CSMs were excluded. Patients with multiple meningiomas and NF2 were excluded, as well as previously irradiated lesions. Histology was available in 46 patients who had undergone microsurgery. The local tumour control rate was 99% at 5 years and 93% at 10 years following SRS. Prior surgery did not influence the outcome, and the authors suggested that those patients with large tumours, or those who present with visual loss generally should undergo microsurgical debulking, followed by adjuvant SRS to achieve tumour control.

Fariselli et al. reported on the efficacy of hypofractionated, stereotactic radiotherapy HSFRT (25 Gy, delivered in 3 to 5 fractions), using Cyberknife™, in order to avoid radiation-induced optic neuropathy (RION—5.1% risk of visual worsening after treatment) while achieving a high PFS rate (93% at 5 years) [57]. Fractionated radiotherapy is also possible with the Gamma Knife model ICON™ (Elekta Instruments, AB, Stockholm, Sweden) [58], which also allows HSFRT for large-volume tumours in patients with major comorbidities and contraindications to surgery, or for tumours in close contact with, or surrounding the optic apparatus.

Although tumour control with primary SRS is good for small- to medium-sized tumours, surgery for these and for larger tumours remains an option for adequately counselled patients who understand the risks and benefits of both treatments [35].

### 4.4. Challenging Aspects Related to SRS

One of the challenges of SRS for CSM remains the proximity of the optic nerves and chiasm. However, the gamma knife has the ability to plan for a low isodose with a very steep dose gradient [59]. A distance of 2–4 mm between the optic apparatus and the target is enough to avoid delivering more than 10 Gy to the optic apparatus [60]. If there is contact between the target volume and the optic pathways, SFRT is recommended [57]. In our series, four patients suffering from optic nerve neuropathy after SRS were treated with am margin dose more than 15 Gy.

Pituitary insufficiency is also a risk, with recent studies showing that a single fraction dose of more than 15 Gy to the pituitary stalk should be avoided [61]. Interestingly, we observed in our series two cases of delayed transitory partial pituitary deficit two years after SRS, which regressed after temporary medication. A margin dose of more than 16 Gy was delivered.

ICA Stenosis after SRS is a potential risk and is well described in the literature [1,2,29,30,34]. We observed five cases of symptomatic ICA stenosis; in one patient the stenosis was remote to the target volume and in three patients, the tumour continued to grow.

### 4.5. Further Aspects of CSM Management

It is now clear that SRS is an effective treatment for CSMs [2,34,62,63,64,65,66,67,68]. The question of the timing of adjuvant SRS after subtotal continues to divide opinion. It is now common practice to monitor residual tumours after surgery using serial MRI scans, and to offer adjuvant radiotherapy or SRS only when growth is confirmed. For vestibular schwannomas [69], subtotal excision followed by SRS as a planned combined approach is an increasingly advocated treatment strategy [70,71]. In our previous report, we commented that “active surveillance” requires ongoing evaluation [24]. Untreated meningiomas, either from the time of diagnosis or after subtotal resection may lead to additional morbidity due to tumour growth [1,2,39,61,72,73,74,75,76,77,78,79,80,81,82,83]. A recent consensus statement on the treatment of CSM from the skull base section of the European Association of Neurological Surgery (EANS) recommends SRS as first-line treatment for small, well defined, pauci-symptomatic lesions in elderly patients, and for larger CSMs not amenable to resection. Microsurgery is recommended for rapidly progressing lesions in young patients presenting with oculomotor, visual, or endocrinological impairments. In order to reduce surgical morbidity, the authors recommend subtotal resection followed by adjuvant SRS in accordance with our outcomes. Conservative treatment with serial imaging follow-up should be proposed in patients with a newly diagnosed asymptomatic CSM that has no mass effect on the adjacent temporal lobe [35].

Complications after SRS are relatively few, but there is a theoretical risk of radiation-induced secondary tumours, or malignant transformation of the treated benign tumour [25,39,56], although this was not seen in our review. Wolf et al. [14] suggest that this risk remains low at long-term follow-up and is similar to the risk of the general population developing primary CNS tumours. 

### 4.6. Limitations of the Study

This outcome study has several limitations. It is retrospective and multicentre, with each unit having slightly different selection criteria, target contouring and dose planning. Nevertheless, the large volume of data collected by the first author was organised in a structured database agreed to by each participating centre, resulting in a uniform data set. There is also potential bias with regard to recurrence/failure rates due to the relatively short follow-up of some patients (24 months). The median time from treatment to tumour growth in our study is four years and nine months (and a mean of five years). Bias is also potentially introduced by loss of patients to follow-up. We elected only to include patients with more than two years of post-treatment imaging, while reducing the clinical follow-up to a minimum of 6 months in order to capture any early adverse treatment effects [24]. As this is a historical study, it includes a range of different planning systems and imaging techniques. All centres treating before 1993 used CT imaging. Between 1994 and 1996, Leksell Gamma Plan software was introduced along with MRI-defined targeting. Since 1998, all treatments were planned with LGP from high-definition MRI images [24].

## 5. Conclusions

We believe SRS is a valuable treatment option for CSM, both as a primary treatment, and as an adjunct following surgery. Imaging diagnosis is reliable in the absence of histology. The risk of pituitary insufficiency is low. Cranial nerve neuropathies improve after SRS in over 20% of patients treated primarily. For those patients best treated surgically, we support the strategy of subtotal microsurgical resection, followed by adjuvant SRS, with or without a period of “active surveillance”, as this clearly reduces surgical morbidity for patients.

## Figures and Tables

**Figure 1 cancers-14-04047-f001:**
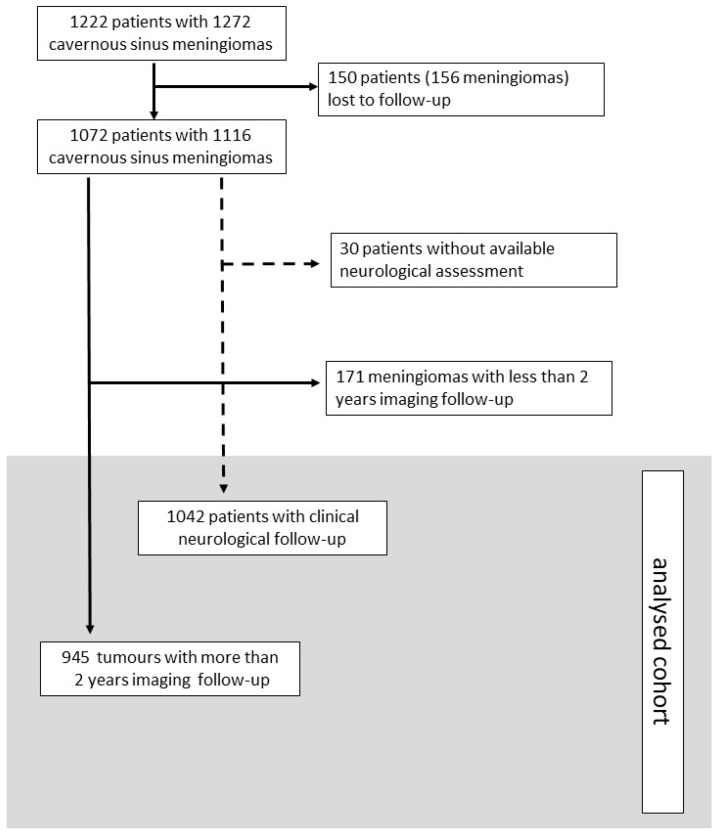
Flow chart of analysed cohort.

**Figure 2 cancers-14-04047-f002:**
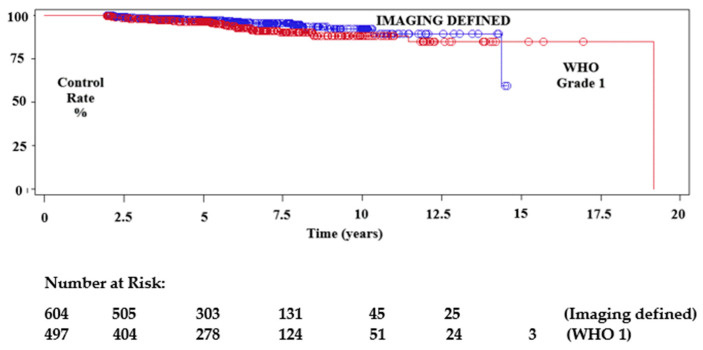
Kaplan–Meier test showing local imaging tumour control after SRS plotted by tumours without histological confirmation as well as histologically confirmed benign meningiomas (WHO Grade 1).

**Figure 3 cancers-14-04047-f003:**
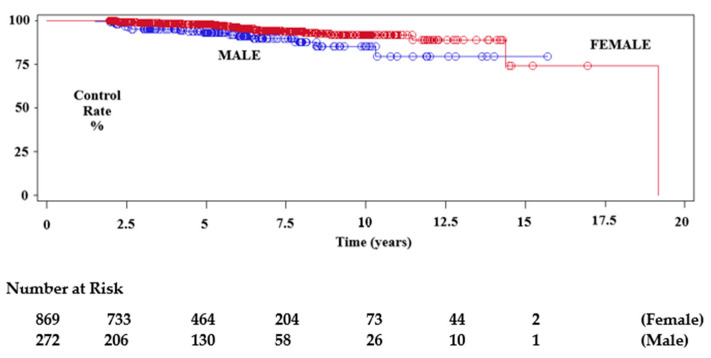
Kaplan–Meier test showing local imaging tumour control after SRS plotted by gender.

**Figure 4 cancers-14-04047-f004:**
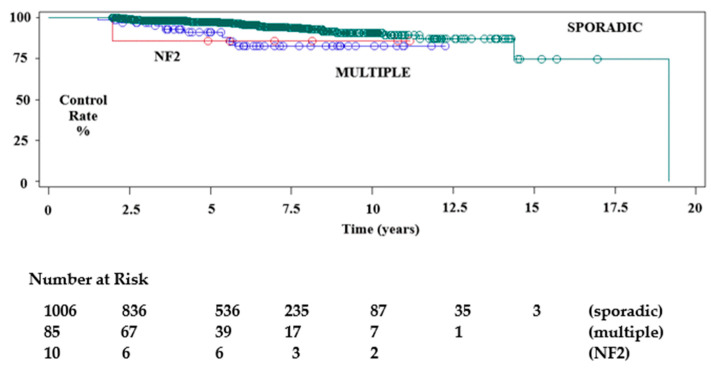
Kaplan–Meier test showing local imaging tumour control after SRS plotted by number of meningiomas treated.

**Figure 5 cancers-14-04047-f005:**
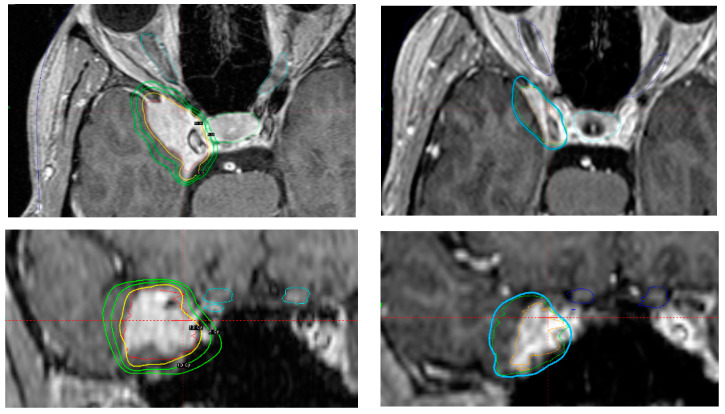
Axial and coronal contrast-enhanced MRI-based treatment (**left
top** and **bottom**) and imaging follow-up 9 years and 6 months after treatment (**right top** and **bottom**). Imaging-defined CSM treated in a 44 year-old woman reporting right periorbital dysesthesia. No further neurological defi cits. No previous surgery. In order to avoid future tumour-associated complications due to the eloquent location, definitive SRS was performed with 13 Gy prescribed to the 65% isodose line. After a follow-up of 114 months, target volume decreased from 5.65 cc to 2.38 cc. The patient developed no new neurological deficits and the periorbital dysesthesia had regressed.

**Table 1 cancers-14-04047-t001:** Characteristics of 1222 patients treated with SRS.

Patient Group	Value
Age at treatment (years) *	55 (±12.0)
Female **	932 (76.3)
Male **	290 (24.7)
Patients with sporadic meningiomas **	1150 (94.1)
Patients with multiple meningiomas ***, **	62 (5.1)
Patients with NF 2 ***, **	10 (0.8)
Neurological Status *****	
Headache **	173 (14.1)
Seizures **	23 (1.8)
Cranial nerve deficit **	973 (79.6)
Hemiparesis, hypoesthesia **	56 (4.6)
Imbalance, ataxia-vertigo **	69 (5.6)
Details of Stereotactic Radiosurgery	
Volume (cm^3^) *, ****	6.3 (±7.0)
Imaging-defined benign meningiomas **	703 (55.2)
Histologically proven WHO Gr. 1 meningiomas **	569 (44.7)
Sporadic meningiomas ***, **	1164 (91.5)
Multiple meningiomas ***, **	95 (7.5)
NF2 meningiomas ***, **	12 (1.0)
Maximal dose (Gy) *	28 (±7.0)
Marginal dose (Gy) *	14.0 (±3.0)
Treatment marginal isodose (%) *	50 (±5.0)
Isocentres *, ****	11 (±9.0)
Dose to optic pathways *	8.0 (±4.0)
Follow-up	
Imaging follow-up (months from treatment) *	61 (±38) 67(±33) *****
Clinical follow-up (months from treatment) *	62 (±39)
Patients lost to follow-up **	134 (10.9%)
Tumours lost to follow-up **	156 (12.2%)
Tumours with follow-up > 5 years **	595 (46.7%)
Tumours with follow-up > 7.5 years **	261 (12.6%)
Tumours with follow-up > 10 years **	103 (8.4%)

* Median and standard deviation. ** Number—(%). *** Multiple meningiomas were defined as more than one tumour treated in the same or different session and not having recurred within the surgical field. All patients diagnosed with neurofibromatosis 2 regardless of tumour-count treatments were analysed separately. NF2-related meningioma was defined as meningioma linked to multiple inherited schwannomas, meningiomas, and ependymomas. **** Volume was available at time of treatment for 1115 tumours and isocentre number for 1148 treatments. ***** The number of symptomatic cases does not correspond to the total number of patients, as some had more than one symptom. A total of 164 patients (13.4%) were asymptomatic.

**Table 2 cancers-14-04047-t002:** Cranial nerve deficits in 1222 patients undergoing SRS for CSMs *.

Neurological Deficit	Number of Patients (%)	
Cranial nerve	Definitive SRS	Adjuvant SRS
I	0 (0%)	1 (0.1%)
II	315 (81%)	294 (51.6%)
III-IV-VI	360 (51%)	261 (45.8%)
V	188 (26.7%)	204(35.8%)
VII	45 (6.4%)	56 (9.8%)
VIII	25 (3.5%)	36 (6.3%)
IX-X-XI	5 (0.7%)	1 (0.1%)
XII	1 (0.1%)	1(0.1%)

* Some patients harboured more than 1 target lesion outside the cavernous sinus. The figures relate to neurological deficits before treatment.

**Table 3 cancers-14-04047-t003:** Progression-free survival rate at 5, 7.5 and 10 years after SRS *.

Variable/Follow-Up Years	5 years	7.5 years	10 years
Imaging Defined Meningiomas	97.2% (95.6–98.4)	95.3% (92.2–97.2)	92.3% (86.8–95.5)
WHO Grade 1 Meningiomas	96.2% (93.6–97.7)	90.1% (85.3–93.3)	87.9% (82.1–91.9)
Female	97.7% (96.1–98.6)	93.6% (90.6–95.7)	91.6% (87.4–94.3)
Male	92.9% (87.7–95.9)	89.6% (82.8–93.7)	85.1% (75.0–91.3)
Sporadic Meningiomas **	97.2% (95.7–98.2)	93.7% (91.0–95.6)	90.8% (86.8–93.6)
Multiple Meningiomas **	90.8% (79.3–96.1)	85.3% (71.1–92.9)	82.4% (67.2–91.0)
NF2 Meningiomas **	85.7% (33.4–97.8)		

** Multiple meningiomas were encoded as one (no) or more than one (yes) meningioma treated. NF2 Meningiomas were analysed separately. * Progression-free survival rate (PFS) and lower-upper confidence limit at 95% (LCL-UCL).

**Table 4 cancers-14-04047-t004:** Univariate analysis of tumour control defined as stable or reduced volume of the tumour irradiated.

**Univariate Chi-Squares for the Log-Rank Test**				
**Variable**	**Chi-Square**		**Prob. > Chi-Square**	
Previous surgery *	2.4020		0.1212	
Gender	7.0674		0.0079	
Multiple Meningiomas **	7.8110		0.0201	
**Unifactorial Cox Proportional Hazard Model**				
**Variable**	**Pr. > Chi Square**	**Hazard Ratio**	**95% Hazard Ratio Confidence Limits**	
Centre	0.0148			
Previous surgery *	0.1242	0.643	(0.366–1.129)	
Age	0.6351	0.994	(0.972–1.018)	
Gender	0.0095	2.149	(1.206–3.830)	
Volume	0.6307	0.988	(0.940–1.038)	
Prescription dose	0.1929	1.056	(0.973–1.145)	
Sporadic tumour vs. Meningiomatosis **	0.0097	2.725	(1.275–5.824)	
Sporadic tumour vs. NF2 **	0.3776	2.446	(0.335–17.851)	

* Previous surgery implies histological confirmation of WHO Grade 1 meningioma. ** Multiple meningiomas were encoded as one (no) or more than one (yes) meningioma treated. NF2 meningiomas were analysed separately.

**Table 5 cancers-14-04047-t005:** Multivariate analysis of tumour control defined as stable or reduced tumour volume on follow-up imaging.

Mutifactorial Cox Regression				
Variable	Pr. > Chi Square	Hazard Ratio	95% Hazard Ratio Confidence Limits	
Centre	0.0160			
Gender	0.0031	2.467	(1.356–4.486)	
Sporadic tumour vs. Meningiomatosis *	0.0056	3.082	(1.389–6.840)	
Sporadic tumour vs. NF2 tumour *	0.6359	1.630	(0.216–12.309)	

* Multiple meningiomas were encoded as one (no) or more than one (yes) meningioma treated. NF2 meningiomas were analysed separately.

**Table 6 cancers-14-04047-t006:** Clinical neurological picture before SRS (left columns) and at last follow-up (right columns) *, **.

Sign-Symptom	Patients with No Symptoms	Patients with Symptoms	Patients with No Symptoms at Last Follow-Up	Patients with Symptoms at Last Follow-Up
Headache	889 85.3%	153 14.7%	940 90.2%	102 9.8%
Cranial nerves deficit	196 18.8%	909 81.2%	340 32.3%	702 67.7%
Hemiplegia Hemiparesis	1003 96.3%	39 3.7%	1006 96.6%	36 3.4%
Dizziness Imbalance Vertigo	993 94.3%	59 5.7%	998 95.4%	54 4.6%
Dysesthesia Hypoesthesia	1030 98.8%	12 1.2%	1030 98.9%	12 1.1%
Seizures ***	1025 98.4%	17 1.6%	1028 98.7%	14 1.3%
Clinical improvement: 460 cases (44.2%) Resolution of symptoms: 241 cases (23.2%) Improvement in cranial nerve neuropathy: 288 cases (27.6%) Resolution of symptoms related to cranial nerve neuropathy: 144 cases (13.5%)				

* Improvement rate is calculated from those patients with a detailed neurological examination (1042 patients, 85.2%). ** Some patients exhibited more than one sign/symptom. *** Epilepsy classification: 0—no symptoms, 1—partial seizure, 2—generalized seizures.

**Table 7 cancers-14-04047-t007:** Complications after SRS in 113 patients *, **.

Sign-Symptom	Mild	Continuous Not Disabling	Continuous Disabling	Temporary	Permanent
Imbalance ataxia Vertigo dizziness	1	1	0	1	1
Vision troubles	3	2	3	0	8
3rd, 4th or 6th nerve palsy	2	13	9	5	19
Trigeminal symptoms	15	14	2	17	14
Facial palsy	2	2	0	3	1
Hearing loss tinnitus	2	2	0	3	1
Symptomatic oedema	2	4	0	5	1
Seizures ***	4	4	---	6	2
Headache	11	10	1	10	12
Hemiplegia Hemiparesis	1	0	0	1	0
Other	0	2	3	4	1
Pituitary deficit	0	2	0	2	0
Permanent mild morbidity rate: 1.13% (15 cases) Permanent continuous (not disabling) morbidity rate: 2.78% (31 cases) Permanent continuous (disabling) morbidity rate: 1.34% (14 cases)					

* Morbidity rate is calculated from those patients with a detailed neurological examination (1042 patients); ** some patients exhibited more than one sign/symptom. *** Epilepsy classification: 0—no symptoms, 1—partial seizure, 2—generalized seizures. Each category was then divided into temporary or permanent sub-groups.

## Data Availability

The data that support the findings of this study are available from the corresponding author, A.S., upon reasonable request.

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
