# Peer review of "Stereotactic Radiosurgery for Benign Cavernous Sinus Meningiomas: A Multicentre Study and Review of the Literature"

_cancers, 2022, doi:10.3390/cancers14164047_

Round 1

Reviewer 1 Report

Well-written manuscript in an important field. No major concerns. Might be better to add 1-2 representative pictures about the tumor and the isodose coverage in the radiation plans. See attached.

Reviewer 2 Report

Major

  1. Please clearly state the aim of the study in the introduction.
  2. Please add references for lines 90-94 and lines 96-99 in the introduction.
  3. The material and methods section it's very confusing. Authors could divide this section into subsections and clearly describe the cohort analyzed, SRS performance, definitions, and stats (please briefly explain what you did in this manuscript). Also, detail the type of analysis performed, and describe the statistical methods used, along with the comparisons performed.
  4. Figure 1 – do the authors mean “1072 patients” or 1078 patients (if only 144 patients were lost to follow-up…)? Please check, and if incorrect rectify.
  5. Line194-197: what is the number of patients to whom the different percentages of meningiomas correspond?
  6. Lines 207-208: which centers provided better tumor control? Any specific reason? Please argue and discuss.
  7. Please always correlate the description of your results with a specific Table(s) and/or Figure(s), both in the Results and the Discussion sections. It's very unsettling to read a paragraph of 5-10 sentences with no reference to a specific "Result", especially in such descriptive work! E.g.: Line194, line197, line 200, line 247, line 249…
  8. Could the authors submit Figures 2, 3, and 4 with higher resolution?
  9. Lines 228-231: what is the purpose of the “Number at Risk”? Does this refer to Figure 4?
  10. Table 6 is not described in the Results.
  11. What are the major significant differences between your clinical study compared to previous studies?
  12. Discussion: Authors described several published works but never related those works with your findings! The main message, the main finding of this work is missing. Authors are more focused on the previous work (reference 20) than on this actual work.
    1. What is discussed in Section 4.1? It is a summary of the description of the results... Authors reported a “poorer control in males, rather females”, how similar/different is the observation compared to other series reports?
    2. Section 4.2: how does the work from Sindou et al. (Reference 12) relate (or not) to the findings here reported? What's the purpose of introducing CSM in this section?
    3. Section 4.3: how does the work describe by Leroy et al. (reference 40) relates (or not) to the findings here reported? Lines 334-335: which results from this manuscript support such evidence? Again, how all the works cited by the authors in this section relates (or not) to the findings here reported? What were the main findings regarding “radiation therapy for CSM”?
    4. Section 4.4: which results support this evidence? Please discuss your results and argue according to the literature. Everything stated in this section was (repeatedly) described before.
    5. Section 4.5: «does your results support the “monitoring by serial imaging after subtotal resection”? (Lines 387-388). Which results support the evidence that “SRV is a safe method for managing benign meningiomas (Lines403-404)?

Minor

  1. Figure 2 - captions are described twice, please rectify.
  2. Define CN (line250)
  3. Please separate the asterisks with a comma (*,**,***).
  4. Define NF2 in Table 1.
  5. Table 4: PFS rate is expressed in % and…?
  6. Table 7: what does it mean * and ** (lines256-237)
  7. Carefully proofread the manuscript to minimize typos, and grammatical errors, maintain writing consistency and avoid redundancy.

E.g.: “tumour vs tumor”; “follow up vs follow-up vs Follow Up”; “NF 2 vs NF2 vs neurofibromatosis type 2 vs type II neurofibromatosis”; usage of abbreviations (why define an abbreviation not to use it); Lines 272-273: “five years vs 67 months.”

  1. Authors’ affiliations start in “22”…order should be re-organized.

Reviewer 3 Report

The study is certainly interesting and valid. The number of enrolled patients is large.
The study only analyzes patients with benign meningiomas. This data should also be specified in the title.
Have the treated patients already undergone surgical treatment? This data should be better clarified in the text.
Have there been any cases of toxicity?

Reviewer 4 Report

Santacroce et al. provide a retrospective review of SRS for cavernous sinus meningiomas.

The paper is of interest and well structured. Its limitations are discussed.

I would only recommend revision by a native English speaker to correct minor errors.

Round 2

Reviewer 1 Report

Excellent work.

Reviewer 2 Report

Dear Antonio Santacroce and colleagues,

 First, accept my apology for my late response.

The revised version addresses the majority of previously raised concerns. I do not have any further questions or concerns. I would like to congratulate you on this (great!) revised form of your manuscript, now acceptable for publication.

I ask the authors to (and once more) carefully proofread your manuscript to minimize typos and maintain writing consistency. E.g.:

  • Line 263 “RS” or “SRS”?
  • Lines 324-325: are the numbers within the brackets referring to the number of patients?
  • Lines 510-515: very confusing
  • “NF2” vs “NF 2”
  • Grade I vs Grade 1
  • starting a sentence as “Five” vs “5”
  • Once an abbreviation is defined used it; for instance, I still find “cavernous sinus meningiomas - CSM” or “ stereotactic radiosurgery -SRS” or “ progression free survival -PFS” along with the text (and captions), rather than their abbreviations.
  • “Follow up” is written in at least 3 different forms. 

Reviewer 3 Report

The authors have fully replied to my observations.